# β-Tricalcium Phosphate-Modified Aerogel Containing PVA/Chitosan Hybrid Nanospun Scaffolds for Bone Regeneration

**DOI:** 10.3390/ijms24087562

**Published:** 2023-04-20

**Authors:** Róbert Boda, István Lázár, Andrea Keczánné-Üveges, József Bakó, Ferenc Tóth, György Trencsényi, Ibolya Kálmán-Szabó, Monika Béresová, Zsófi Sajtos, Etelka D. Tóth, Ádám Deák, Adrienn Tóth, Dóra Horváth, Botond Gaál, Lajos Daróczi, Balázs Dezső, László Ducza, Csaba Hegedűs

**Affiliations:** 1Department of Oral and Maxillofacial Surgery, Faculty of Dentistry, University of Debrecen, 4032 Debrecen, Hungary; 2Department of Inorganic and Analytical Chemistry, Faculty of Science and Technology, University of Debrecen, 4032 Debrecen, Hungary; 3Department of Biomaterials and Prosthetic Dentistry, Faculty of Dentistry, University of Debrecen, 4032 Debrecen, Hungary; 4Division of Nuclear Medicine and Translational Imaging, Department of Medical Imaging, Faculty of Medicine, University of Debrecen, 4032 Debrecen, Hungary; 5Department of Medical Imaging, Faculty of Medicine, University of Debrecen, 4032 Debrecen, Hungary; 6Department of Dentoalveolar Surgery, University of Debrecen, 4032 Debrecen, Hungary; 7Department of Operative Techniques and Surgical Research, Faculty of Medicine, University of Debrecen, 4032 Debrecen, Hungary; 8Department of Anatomy, Histology and Embryology, Faculty of Medicine, University of Debrecen, 4032 Debrecen, Hungary; 9Department of Solid State Physics, University of Debrecen, 4002 Debrecen, Hungary; 10Department of Oral Pathology and Microbiology, Faculty of Dentistry, University of Debrecen, 4032 Debrecen, Hungary

**Keywords:** aerogel, β-tricalcium phosphate, electrospinning, electrospun meshes, tissue engineering

## Abstract

Electrospinning has recently been recognized as a potential method for use in biomedical applications such as nanofiber-based drug delivery or tissue engineering scaffolds. The present study aimed to demonstrate the electrospinning preparation and suitability of β-tricalcium phosphate-modified aerogel containing polyvinyl alcohol/chitosan fibrous meshes (BTCP-AE-FMs) for bone regeneration under in vitro and in vivo conditions. The mesh physicochemical properties included a 147 ± 50 nm fibrous structure, in aqueous media the contact angles were 64.1 ± 1.7°, and it released Ca, P, and Si. The viability of dental pulp stem cells on the BTCP-AE-FM was proven by an alamarBlue assay and with a scanning electron microscope. Critical-size calvarial defects in rats were performed as in vivo experiments to investigate the influence of meshes on bone regeneration. PET imaging using ^18^F-sodium fluoride standardized uptake values (SUVs) detected 7.40 ± 1.03 using polyvinyl alcohol/chitosan fibrous meshes (FMs) while 10.72 ± 1.11 with BTCP-AE-FMs after 6 months. New bone formations were confirmed by histological analysis. Despite a slight change in the morphology of the mesh because of cross-linking, the BTCP-AE-FM basically retained its fibrous, porous structure and hydrophilic and biocompatible character. Our experiments proved that hybrid nanospun scaffold composite mesh could be a new experimental bone substitute bioactive material in future medical practice.

## 1. Introduction

Facial bone defects treated in dental practice originate from numerous causes, including traumas, congenital malformations, or the resection of a neoplasm. Until recently, the replacement of bone deficits has been a major challenge in patient care. Several conventional solutions are known to compensate for the lack of bone (autograft, isograft, allograft, xenograft, alloplast, composite graft), during which the vast majority of the materials used are biodegradable and possess mechanical properties close to human bone. In the case of some bone deficiency treatments, such as the restoration and maintenance of bone function, the resistance of the implanted scaffold to high mechanical stress is not a basic requirement, as it is primarily not used to replace directly the lost bone tissue, but to create a microenvironment that supports bone regeneration processes. This type of scaffold, in contrast to bone replacement materials exposed to high strain (such as magnesium scaffold [1], can be gels, thin films, or fibrous meshes.

Nowadays, growing attention is paid in dentistry to biomimetic fibrous meshes produced via the electrospinning method. The merits of the electrospinning technique include controllability of fiber diameters, variability in methods, and possible application of different materials. These materials can enhance cell attachment, growth, and proliferation [2,3]. Polymers combined with antibacterial agents can enhance wound healing and control cytotoxicity side effects [4]. The optimal size of pores and the permeability of the different scaffolds are important factors for different ions, signaling molecules, and cells [5].

Enhanced hybrid nanofibrous meshes can be produced by blending biopolymers (e.g., gelatin, silk fibroin, collagen, chitosan (Ch)) and synthetic polymers (e.g., poly(vinyl alcohol) (PVA), poly(lactic acid), polyvinyl pyrrolidone, polycaprolactone) wherein the advantages of each macromolecule can be combined. For instance, PVA/Ch polymer blends, in addition to their biodegradable and biocompatible properties, can play a significant role in cell–cell interactions because of the structural similarity of chitosan to the glycosaminoglycans, and they possess a flexible and hydrophilic character because of the PVA. Furthermore, the PVA and Ch can form hydrogen bonds with each other, thus ensuring the homogeneity of the mixed polymer solution for the electrospinning process [6].

Besides the qualitative properties and molar ratios of the polymers, the parameter settings of the electrospinning device and the post-processing modifications (such as plasma and laser treatment, surface functionalization, inorganic combination, and cross-linking method) also have a significant effect on the physicochemical properties of the fibrous meshes [7,8]. In our previous research, a porous, partially cross-linked, hydrophilic PVA/Ch blend mesh with a nanofibrous structure was produced by optimizing the polymer composition and the cross-linking with citric acid [9].

Several types of meshes have been modified e.g., with biomolecules, bioceramics. These molecules’ release can be tuned by adjusting scaffold porosity and composition. The different molecules and inorganic components (e.g., growth factor, bioglass, Si, Mg, hydroxyapatite) can be loaded into the meshes by adsorption, covalent binding, or mixing to the electrospinning solution [10,11,12].

Autologous bone is the gold standard for bone replacement, while β-tricalcium phosphate (BTCP) is one of the most commonly used materials among alloplasts. BTCP-regulated osteogenesis is the likely mechanism by which BTCP can directly or indirectly (through the release of ionic components such as Ca^2+^ and PO_4_^3−^) promote vascular development, regulate the release of growth factors, alter blood clots, and promote osteoblast differentiation. Based on the results, Ca^2+^-mediated osteoblast differentiation is achieved through the activation of calmodulin and calmodulin-dependent kinase II (CaM-CaMKII) pathways [13,14].

Silicon is an important factor in bone formation, and can play an essential role in bone metabolism. The orthosilicic acid can increase the expression of BMP-2, and the BMP-2/Smad1/5/RUNX2 signaling pathway participates in the silicon-mediated induction of COL-1 and osteocalcin synthesis [15].

In our previous papers, the solubility, ionic equilibria, cell viability, proliferation, biological activity, and influence on in vivo tissue regeneration of a new artificial bone substitute material, silica aerogel–-β-tricalcium phosphate (BTCP-AE) nanocomposite, and the preparation of PVA/Ch nanofibrous structures were presented [9,16]. The advantage of the amorphous silica aerogel matrix is that the extremely high surface area and higher solubility provide a slow and continuous release of the orthosilicate ions, which promotes Type I collagen formation. The guest material, BTCP, is widely used in medical practice and its properties are well documented (Calcium orthophosphate (Ca_3_PO_4_)-based bone graft) [17].

In the present study, we combined a PVA/Ch polymer blend with BTCP-AE to form a nanofibrous polymer composite mesh, which provides Ca-, P-, and Si-leachable components for new bone formation to improve the healing process.

This study aimed to demonstrate the applicability of BTCP-modified aerogel-containing PVA/chitosan hybrid nanospun scaffolds for bone regeneration in both in vitro and in vivo experimental approaches.

## 2. Results

### 2.1. Morphological Characterization and Physicochemical Properties of Electrospun FMs and BTCP-AE-FMs

Meshes made with electrospinning had a reticular structure constructed of thin, fine fibers. The mean thicknesses of the non-woven FM and BTCP-AE-FM were 210 ± 36 and 150 ± 21 µm, respectively. An electrospun disk-shaped mesh, folded in half, is demonstrated in Figure 1.

Using SEM, the spatial distribution and diameter of the trabeculae in the non-cross-linked and cross-linked FMs as well as the BTCP-AE-FMs were compared. Considering the fiber diameters and additional surface provided in the cross-linked meshes, they offered promising biomedical properties (Figure 2). The aqueous solutions of PVA and Ch mixed well into a homogeneous solution, providing homogeneous PVA/Ch blend fibers when dried during electrospinning.

The SEM image in Figure 2a shows that the non-cross-linked FM contained uniform, smooth, cylindrical nanofibers. However, during cross-linking, the structure of the mesh was significantly altered; the cylindrical structure of the nanofibers disappeared, the fibers flattened, or the fibers were interconnected at the contact cross-points and in some cases along the entire length of the fibers. The cross-linked structure, formed by inter-fiber chemical bonds, resulted in the integration of the fibers and the reduction in porosity of the original skeletal structure (Figure 2b). The SEM image of the cross-linked BTCP-AE-FM (Figure 2c) shows fibers that were less uniform than the non-cross-linked FM but more uniform than the cross-linked FM. This suggested that BTCP-AE used as an inorganic additive affected both fiber formation and cross-linking. The jet instability due to the BTCP-AE particle content was the reason the electrospun fiber sizes were different. In addition, the resulting fibers had a smooth surface, with perceptible grooves in the surface layer because the inorganic particles did not cause faster evaporating solvent-rich areas in the polymer solution. The formation of inter-fiber chemical bonds during cross-linking was partially inhibited by the inorganic filler, which resulted in less chain entanglement. The average fiber diameters (AFDs) of the non-cross-linked FM, cross-linked FM, and cross-linked BTCP-AE-FM were 107, 185, and 147 nm, respectively. We concluded that cross-linking of the FM resulted in an increase in the average diameter through chain entanglement (Figure 2e). The inorganic additive in the BTCP-AE-FM modified the viscosity and conductivity of the polymer solution, whose synergy affected the diameter of the electrospun fibers and the cross-linking reaction. In addition to a slight increase in the average fiber diameter, a widening of the diameter distribution was observed (Figure 2f).

The β-tricalcium phosphate–mesoporous silica aerogel composite (BTCP-AE) powder as the inorganic additive and the BTCP-AE-FM were measured by SEM (Figure 3a,c) and completed with analysis by energy dispersive X-ray spectroscopy (EDS) (Figure 3b,d). In some instances, BTCP-AE inorganic composite particles of the BTCP-AE-FMs were attached to the PVA/Ch polymer fibers between the fibers or on the mesh (Figure 3c). The EDS spectrum of a typical particle in the BTCP-AE-FM is presented in Figure 3d. The EDS analysis confirmed that the main elements on the surface of the particle were C, O, Ca, Si, and P. The EDS analysis was used only to confirm that the typical particles in the mesh were BTCP-AE which was supported by the presence of the Si, Ca, and P elements.

Since nanofibrous composites are not ideally flat—their surface bears roughnesses and air-trapping pores—the contact angle measurement can provide additional information for the comparative study of fibrous materials with a similar structure. Figure 4 shows the contact angles of the BTCP-AE-FM before and after cross-linking, indicating that cross-linking increased the contact angle from 46.5° to 64.1°. Contact angles of cross-linked meshes were still less than 90°, meaning that the cross-linked materials retained their hydrophilic character.

The chemical cross-linking reaction of polyvinyl alcohol, chitosan, and citric acid as well as the IR analysis of the initial and cross-linked materials are discussed in detail in the literature [18,19,20,21,22]. Although the IR spectra of some of the starting materials can be downloaded from a spectral database [23], their quality, recording mode, and material origin are variable; therefore, all initial commodities and intermediates were also recorded for the sake of comparability. Here, the analysis and computer fitting of the ATR-IR data were focused on the characteristic spectral differences, which indicated a shift in the chemical bonding structure. The IR spectra of the BTCP-AE-FM and the aerogel-free PVA/Ch fibrous meshes were very similar. The spectra of both materials were dominated by the spectrum of polyvinyl alcohol. Because of their lower concentration, chitosan and citric acid contributed less intensely to the final spectrum. The high-wavenumber region of the spectra showed two broad peaks. The first one belonged to the overlapping stretching frequencies of ν(O-H) and ν(N-H) with the maximum at 3318 cm^−1^. Although ν(N-H) peaks were resolved in the solid chitosan (Figure 5b), they could not be observed separately because of protonation with acetic and citric acid, and the chemical reaction occurred on cross-linking. Stretching of the carbon–hydrogen bonds in both polyvinyl alcohol and chitosan showed two maxima at 2940 and 2913 cm^−1^, with a shoulder at 2874 cm^−1^. The first two were characteristic of PVA, while the shoulder could be observed in both PVA and chitosan. The shallower nature of the valley in between the peaks at 3318 and 2940 cm^−1^ might indicate the presence of chitosan. Two resonances indicated characteristic changes in the structure of the BTCP-AE-FM compared to the FM. A very obvious mark of a chemical reaction was the appearance of the new amide-II peak at 1585 cm^−1^ (blue arrow, Figure 5a). Simultaneously, the intensity drop of the shoulder belonging to the ester ν(C-O) stretching frequency at 1200 cm^−1^ (blue arrow, Figure 6) was noted. Both were related to the presence of aerogel particles in the polymeric matrix. High-surface-area silica aerogels are strong dehydrating agents, especially at the cross-linking temperature. In the PVA/Ch matrix, finely dispersed aerogel particles promoted the condensation reaction of chitosan amino groups with citric acid, forming amide groups that resulted in the appearance of the amide-II peak. In accordance with the change, the number of ester groups formed from the hydroxyl groups of PVA and chitosan molecules with citric acid decreased; thus, the intensity of the ester ν(C-O) stretching peak decreased (Figure 5). Characteristic IR frequencies and their assignation are given in Table 1.

The increasing intensity in the carbonyl stretching frequency (as compared with the intensity of the wagging deformation frequency γ_w_(CH) of C-H bonds at 1239 cm^−1^) was also in conjunction with the reduced number of ester groups. Although this change was indicative, quantitative calculations could not be performed because of the formation of an amide-I peak, as well as the presence of the residual ester groups in the starting material PVA and the remaining carboxylic groups of citric acid after cross-linking.

The silica aerogel composite BTCP-AE embedded in the BTCP-AE-FM could not be observed directly in the spectrum. However, its presence was mirrored in the changes in peak intensities in the 700–1500 cm^−1^ region. The silica aerogel composite showed a strong absorption around 1100 cm^−1^ (Figure 5b). Although the spectrum was poor in peaks, the intensity changes were used to initiate a calculation in which the linear combination of the spectra of the FM, BTCP-AE, and CA was fitted to the spectrum of the BTCP-AE-FM by the least squares method using the Solver module of Excel for the 700–1500 cm^−1^ region. The original and the fitted spectra (Calcd. spectrum), as well as the components in the fitted ratio, are shown in Figure 6. Other combinations with pristine silica aerogel, β-tricalcium phosphate, and citric acid were also tested, but neither of them showed a considerable difference. Even the combination of citric acid with BTCP-AE and the FM did not cause any significant change. According to the calculations, the residual concentration of citric acid in the BTCP-AE-FM was almost negligible, indicating that nearly the entire amount was used up in the cross-linking reaction.

### 2.2. Leachable Part of the BTCP-AE-FM and Ca, P, and Si Releasing

The solubility of the mesh affects the release of the bioactive components, so the release behavior and weight loss were also measured. During the one-week investigation, the weight losses at different times resulted in values between 10% and 30% (Figure 7a). The weight loss in the early stage of degradation (1 h) was about 20%, and its definite increase was not observed as a function of time. Our results confirmed that the rare, cross-linked structure enabled rapid diffusion of macromolecules not connected by intermolecular chemical bonds into water. The release behavior of the Ca, P, and Si ions was consistent with the dissolution behavior of the BTCP-AE-FM (Figure 7b). The rapid dissolution of macromolecules in the early stages facilitated the rapid release of bioactive inorganic components. The Ca/P ratio calculated from releasing data of Ca and P was about 1.5, which corresponded to the theoretical Ca/P ratio and was close to the natural bone Ca/P molar ratio (1.667) (Figure 7c).

### 2.3. In Vitro Evaluation

The viability of DPSC cells grown in the presence (CM+ and OM+) or absence (CM and OM) of AE-fiber samples was estimated using alamarBlue assay after 7 and 14 days to inspect the biocompatibility of the samples. We observed that after 14 days, however, the viability of the cells grown in OM or OM+ was significantly higher compared with that in CM (Figure 8) or CM+, and the presence of the AE-fiber samples in CM+ or OM+ did not affect the viability of the cells compared with that of CM or OM, respectively.

The live–dead assay (Figure 9) confirmed that the ratio of live and dead cells did not alter in the presence of the BTCP-AE-FM samples. Unchanged viability of DPSCs in the presence of the BTCP-AE-FM samples compared to the absence of samples proved that the meshes had no cytotoxic effect on the DPSCs, suggesting that the utilization of meshes in living organisms would act similarly.

SEM images (Figure 10) confirmed that DPSCs were able to grow on the surface and partly inside the fiber net. The cells took a slightly elongated shape, which was somewhat similar to those observed on coherent surfaces, suggesting that the fiber net can provide an ideal environment for cell adhesion.

### 2.4. MicroCT

Micro-CT reconstructions confirmed that in the case of the FM implants, bone regrowth into the defect was delayed nearly 1–2 months as compared with when the BTCP-AE-FMs were applied (Figure 11). With both meshes applied, it was generally noted that bone repair was uneven around the trepanned perimeter. With micro-CT analysis, the osteoid or novel bone coverage of the defect appeared approx. 16% after 1 month, 34% after 3 months, and 60% at 6 months of survival with the FM. With the BTCP-AE-FM, coverage was 31%, 55%, and 94% at the respective months of survival.

### 2.5. PET Imaging Using [^18^F]fluoride

For the in vivo imaging of osteoblast activity at the site of cranial trepanation, experimental animals were injected with [^18^F]fluoride, and PET images were obtained 1, 3, and 6 months after the implantation of the FM (Figure 12A–C) or TCP-AE-FM (Figure 12D–F). Through qualitative analysis of the decay-corrected PET images, we found that the osteoblast activity and the healing process could be followed using [^18^F]fluoride from 1 month to 6 months. However, a difference was recognized between the two scaffolds 3 months after surgery, according to which earlier healing was observed with the use of the BTCP-AE-FM with higher radiopharmaceutical accumulation, and this increased [^18^F]fluoride uptake was also verified at 6 months. These visual observations were confirmed by the quantitative SUV analysis, where increasing SUV_mean_ values were registered from 1 to 6 months using both scaffolds. The SUV_mean_ values of the BTCP-AE-FM implant-bearing rats were relatively higher than those of the FM scaffold-bearing animals at each investigated time point (Table 2).

### 2.6. Histological Evaluation

The periosteum was repaired in a short time after surgery on both facets of the bone plates, as we found them nearly intact even after the first postoperative month. In none of the sections did we notice signs of acute or chronic inflammatory response; in fact, macrophages, neutrophils, or lymphocytes were generally sparse throughout the healing sites. Surprisingly, the proximity of both fibrous composite meshes was found devoid of phagocytes. Biodegradation of the scaffolds, therefore, followed a presumably slow/very slow pace since its thickness remained nearly the same until the postoperative sixth month, in the case of both the FM and BTCP-AE-FM (Figure 13D,F). Osteoclast activity was also generally absent by even in the first month and after. In the first postoperative month, a cone-shaped newly forming woven bone, the osteogenic front, protruded into the defected space (Figure 13A,B,E), fitted firmly to the trepan cut edges. In small numbers and sizes, paler stained bony islets were also noticed within the defective area. Both sites represented intensive osteogenesis, for they were surrounded by osteoblasts. Among osteocytes, the bone matrix showed a rather stratified structure because of appositional matrix build-up. Despite the early onset of osteogenesis, most of the defects at this stage remained sealed with collagenous tissue containing many fibrocytes, although its eosinophilia was pale. Fibroblasts and collagen strands both aligned with the FM and BTCP-AE-FM scaffolds. Interrupting the fibrous tissue, numerous endothelium-lined oval lumens represented intensive neovascularization throughout the healing defect. Usually in the middle of the defect, small areas of mesenchyme-like zones were also spotted. In the third postoperative month, osteogenic fronts progressed centripetally, and their interface with the original calvaria was no longer visible. The rate of osteogenesis, however, seemed heterogeneous around the trepanned perimeter, detected also by radiographic imaging. The defect was still filled by mainly collagenous tissue; nevertheless, the BTCP-AE-FM implantation resulted in a thicker closure after 3 months, in comparison with the native membrane (Figure 13C,F). Mesenchymal zones disappeared. In the sixth postoperative month, bone tissue predominated over the collagenous tissues in the defect. Bone islets and peripheral osteogenic fronts overlapped and fused. In the case of the BTCP-AE-FM, the bone-to-membrane connection appeared very intimate, where the scaffold was engulfed with new bone tissue (Figure 13D,G).

## 3. Discussion

Bone tissue is one of the primary supportive tissues of the body, possibly considered a biocomposite or nanocomposite, consisting of an organic (mostly collagen proteins) and inorganic matrix (hydroxyapatite). Nowadays, tissue engineering is a highly researched scientific field, creating the opportunity to recover lost physiological functions through the production of novel, complex systems. In our studies, we hypothesized that the mentioned properties can be achieved by BTCP-modified aerogel containing PVA/Ch hybrid fibrous scaffolds produced by electrospinning for bone regeneration.

Because of its beneficial biological properties, such as biodegradability, biocompatibility, and antibacterial activity, chitosan is a widely used biomedical polymer [24] and, thus, is also a frequently used raw material for electrospun meshes [25,26]. PVA, the other compound in the bicomponent matrix of meshes, is also a biodegradable, biocompatible polymer, with the additional advantage of excellent water solubility [27]. In addition to its chemical, physical, and biological properties, the structural characteristics of the scaffold can play a significant role in biomedical applications [28]. The morphological properties of BTCP-AE-FMs produced by electrospinning were examined with low-vacuum SEM, and the fiber diameter distribution was determined by ImageJ software. As demonstrated by our results, the approximately 210 µm thick BTCP-AE-FMs materials are made up of fibers with a 147 ± 50 nm AFD and random orientation, in the diameter range from 50 to 450 nm. By SEM, it was possible to detect differences between the morphological properties of cross-linked FM and cross-linked BTCP-AE-FM that could be related to the inhibition effect of BTCP-AE inorganic composite to the inter-fiber cross-linking reaction. However, cross-linking reactions could also be inside the composite fibers, where the dispersed aerogel particles promoted the reactions of Ch amino groups with citric acid as is shown in the IR spectrum. It means that the presence of the BTCP-AE particles in the fiber is favorable to intra-fiber cross-linking reactions, which is demonstrated by the SEM through the remaining porous structure and the narrower fibers than the cross-linked FM. Sisson et al. found that cell infiltration of MG63 cells was more efficient in the large-diameter (600 ± 110 nm) gelatin scaffold than in the small-diameter one (110 ± 40 nm) [29]. Fibers with a diameter of 400–450 nm in the BTCP-AE-FM promote cellular infiltration; thus, the scaffold can affect cell attachment, spreading, proliferation, and differentiation. According to Osorio-Arciniega et al., the critical parameter of the scaffold such as physical, chemical, and surface topography properties can induce cellular recognition signals. Their results showed well attachment of hFOB cells to the polylactic acid/zirconium oxide fiber composite with very spreading, elongated, and lamellae morphology [30]. Our study found that the morphology of the BTCP-AE-FM nanocomposite demonstrated an excellent surface for the attachment of DPSCs as is shown in an SEM image. In addition, the retained hydrophilic nature of the cross-linked fibrous meshes (CA = 64.1°) and the special microenvironment created by BTCP-AE-FM in an aqueous medium with the releasing Ca, P, and Si can also induce cell adhesion. Furthermore, these could be improved by the alignment of fibers was shown in the study of Xie et al. [31]. These aligned nanofibers have an effect on the cell morphology, promote cell migration, and significantly improve cell growth.

Regarding the in vivo applicability of a β-tricalcium phosphate-modified aerogel containing PVA/chitosan fibrous meshes for guiding re-ossification in a bone tissue defect, the present study demonstrated the powerful osteo-inductive feature of our BTCP-AE-FM construct to facilitate bone healing in a rat calvaria critical size model. The grade of ossification appeared comparable with our earlier in vivo results when mesoporous BTCP-AE scaffolds were used for bone defect repair of up to 6 months duration [16]. Accordingly, both showed a sub-total (approximately 80%) ossification of the calvaria bone defect by this time. Remarkably, however, the use of nano-sized compounds and the significantly lower BTCP-AE concentration in the BTCP-AE-FM for creating mesh films to cover the bone defects appeared to prevent chronic persistent inflammation during the late phase of healing, as compared with our previous results with mesoporous BTCP-AE (Figure 14), while the extent of re-ossification remained the same. Supposedly, the low amount of the residual BTCP-AE-FM inside the repairing bone defect, which normally would include inflammation, might explain the absence of acute and chronic foreign body giant cell reaction.

Certain electrospun mesh composites were similarly tested for osteoinduction. Their conclusions were that mesenchyme to osteoblast transformations are promoted with filled meshes [33,34]. We noticed that, regardless of the filling material and post-operative time point, the defect is filled with a highly heterogenous tissue, until its entire closure. The new tissue mass contained the osteogenic fronts and later also islets (Figure 13A), ventral- and dorsal periosteum, an irregular collagenous network, various forms of vascularization, and, incidentally in the early postoperative time, mesenchyme-like tissue matter. Both the osteogenic front and bone islets establish compact bone, whose internal structure appears stratified with HE staining, meaning that deeper parts are stained pale, whereas superficial layers exceed in eosinophilia. We suspect the differing state of matrix mineralization in its background, during the progress of appositional bone regrowth. One major question of the present Calvaria model is the cellular origin of osteogenic centers. Earlier works [35,36] suggested three possible sources: (i) from the periosteum of the original calvaria (ii) bone marrow-derived undifferentiated cells and (iii) pericytes of dural blood vessels [37]. The periosteal origin still temps to be evident, concluded from the rapid repair of the periosteum on both ventral- and dorsal surfaces; furthermore, the osteogenic front firmly fitted with the trepanned edges. Osteogenic islets most probably derive from either periosteal osteoprogenitors or bone-marrow-derived stem cells, arriving via the initial perioperative hematoma, or sinusoid blood vessels. Direct transposition of dural pericytes cannot be excluded either, as the ventral periosteum does have a certain incomplete period. Nevertheless, the conclusion is that the reossification of the 8 mm critical calvaria defect is most probably multifocal, to where osteogenic differentiating cells infiltrate from various origins.

Using ^18^F-sodium fluoride ([^18^F]fluoride) radiopharmaceutical, positron emission tomography (PET) plays an important role in the early diagnosis of bone disorders and monitoring of the treatment response. As [^18^F]fluoride is highly specific for the osteoblastic activity in osseous tissues, this radiotracer is ideal for the in vivo PET imaging and monitoring of physiological and pathological osseous conditions [38]. In the current study, [^18^F]fluoride PET imaging was used for the longitudinal assessment of the healing of the skull after the implantation of the FM and BTCP-AE-FM. In PET imaging, [^18^F]fluoride sensitively displayed the metabolic activity of the bones [39,40]; hence, osteoblast activity induced by the implants could also be monitored during our experiments.

Based on the presented data, we concluded that native fibers could accelerate bone regeneration; furthermore, our results demonstrated the high potential of the BTCP-AE-FM composite material for bone tissue engineering. The limitation of this electrospinning method does not allow a higher amount of BTCP-AE inorganic composite embedded into the feed material. The further aim is to improve the biophysical properties of the composite mesh using different inorganic components and additives. For the optimization of some parameters, e.g., fiber diameter distribution, and creation of mesh density, the in silico model may be more suitable than the experimental animal test method. This model facilitates internal parameter design and could improve the scaffold usability for bone tissue engineering [41,42].

## 4. Materials and Methods

### 4.1. Sample Preparation

In our previous research, several polymeric and cross-linking agent concentrations (PVA/Ch/citric acid ratio) and electrospinning parameters were applied to investigate their effect on the hydrophilicity and cell viability of the meshes [9]. We achieved appropriate hydrophilicity of the mesh with the optimized chemical composition and cross-linking. Based on these optimized parameters, PVA/Ch meshes were prepared, and inorganic composite modified PVA/Ch meshes were improved. Polyvinyl alcohol (Merck, Darmstadt, Germany) (65 kDa, polydispersity: 3.168) and chitosan (Sigma-Aldrich, St. Louis, MO, USA) (750–1000 kDa) blended fibers were created under optimized dissolution and electrospinning parameters. A 10% *w/w* solution of PVA was prepared by the addition of water to the polymer, and a 4% *w/w* solution of Ch was prepared by the addition of acetic acid (2% *v/v*) to the solid Ch. The solutions were stirred for 24 h at 80 °C. The monolithic BTCP-AE inorganic nanocomposite was ground with Analysette 3 Vibratory Sieve Shaker (Fritch, Oberstein, Germany) at 5 min 2.0 mm amplitude. Zirconium balls were used combined with Zirconia lined milling bowl. Calculated amounts of PVA solution (6.6% PVA content in the final mixture) and Ch solution (1.3% Ch content in the final mixture) were mixed, then the ground BTCP-AE (1%) was added and suspended into it; finally, the citric acid (2.9%) was stirred into the mixture. β-tricalcium phosphate–mesoporous silica aerogel composite (BTCP-AE) was made at 800 °C for 1 h, as earlier described by [43]. Polyvinyl alcohol and chitosan blended fibrous meshes (FMs) and inorganic composite modified PVA and Ch hybrid fibrous composite meshes (BTCP-AE-FMs) were also prepared.

The PVA/Ch solution or BTCP-AE/PVA/Ch suspension was placed in a plastic 5 mL tube and fibers were developed by the electrospinning device (Nanospinner NS1, Inovenso Ltd., Istanbul, Turkey) under 25 kV in a 13 cm tip to collector distance at a flow rate of 0.62 mL/h. The PVA/Ch solution and the BTCP-AE/PVA/Ch suspension were electrospun for 1 h at the conditions mentioned above. To enhance the physical and biological features of the mesh, it was cross-linked with citric acid (VWR Chemicals, Leuven, Belgium) at 120 °C for 4 h. The workflow of the fabrication and characterization of the chemically cross-linked biodegradable scaffolds can be seen in Figure 15.

### 4.2. Infrared Spectroscopy

Infrared spectroscopy was performed using the Smart iTR™ Attenuated Total Reflection (ATR) technique. A Nicolet 6700 FT-IR Spectrometer (Thermo Fisher Scientific, Waltham, MA, USA) fitted with a diamond ATR crystal was used. Samples were scanned 16 times at 0.482 cm^−1^ resolution. Before spectral evaluation, ATR correction was performed.

### 4.3. Scanning Electron Microscopy (SEM)

The surface morphology of the electrospun samples was characterized with a dual-beam scanning electron microscope (Thermo Fisher Scientific Scios2; FIB-SEM, Waltham, MA, USA). To avoid charge accumulation of non-conductive samples, observations were made at a low accelerating voltage of 2 kV.

### 4.4. Hydrophilicity of BTCP-AE-FMs

The hydrophilicity of the BTCP-AE-FMs was measured using a water contact angle analyzer (Krüss GmbH, Hamburg, Germany) at 20 °C, applying the Young–Laplace model. A droplet of water (2 µL) was placed vertically onto the surface of the scaffold, and the contact angle was measured (Figure 16) in the first 5 s after placing the water drop, as soon as the water droplet completely touched the membrane. Measurements on triplicate samples were repeated 5×.

### 4.5. Leachable Part of the BTCP-AE-FM and Ca, P, and Si Releasing

Disk-shaped BTCP-AE-FM specimens (8 mm, n = 3) were dry stored at 37 °C for 24 h. After measuring their weight (w_0_), the samples were immersed in 5 mL purified water, incubated at 37 °C, and stirred at 100 rpm. The weight of the scaffolds and volume of the water ratio was 0.5 mg/mL. Inductively coupled plasma atomic emission spectrometry (ICP-OES 5110 Vertical Dual View, Agilent Technologies, Santa Clara, CA, USA) was used to detect the presence of Ca, P, and Si in the solution after 10 min, 30 min, 1 h, 4 h, 8 h, 1 day, 3 days, and 7 days of sample soaking. An autosampler (Agilent SPS4), Meinhard^®^ type nebulizer, and double pass spray chamber were used. Standard solutions were prepared from the mono-element spectroscopic standard of 1000 mg L^−1^ stock solution (ICP IV, Merck). A 5-point calibration was used, for which standard solutions were diluted in ultrapure water containing 0.1 M HNO_3_. Before the ICP-OES analysis, the samples were filtered (pore size: 0.4 µm) and acidified with 100 µL cc. nitric acid. In parallel with the releasing test, a simplified solubility test in water was performed to quantify the water-soluble content. The rinsed and dried (at 20 °C for 24 h) BTCP-AE-FMs were weighted (w_t_) at the predetermined intervals. The following equation was used for calculating weight loss:$$\mathrm{Weight}\;\mathrm{Loss}\;\left[\%\right]=\frac{W_{0}-W_{t}}{W_{o}}\cdot 100$$

### 4.6. Cell Culturing and Viability Assays

Dental pulp stem cells (DPSCs) were cultured in DMEM-F12 medium (Thermo Fisher Scientific, Waltham, MA, USA) supplemented with 10 *v/v*% fetal bovine serum, 1 *v/v*% GlutaMAX, and 1 m/m% antibiotic–antimycotic (all from Thermo Fisher Scientific); this medium is referred to in the text as the control medium (CM). The osteoinductive medium (OM) was prepared by supplementing the CM with 10 mM β-glycerolphosphate, 50 μg/mL ascorbic acid, 0.1 μM dexamethasone, and 50 nM vitamin D3 (all from Sigma Aldrich, St. Louis, MO, USA).

The cells were seeded into 12-well plates (Thermo Fisher Scientific, Rosklide, Denmark) in a concentration of 10^5^/well and incubated in the CM under the same conditions until the next day. On day 0, the media were replaced with fresh CM or OM, and inserts with BTCP-AE-FMs were placed into the wells. The BTCP-AE-FM samples were first sterilized by in situ-generated O_3_ (OzoneDTA O_3_ generator, Apoza Enterprise, Taiwan, ROC) and then by 30 min UV light/side. Wells with the CM or OM but without the inserts, and BTCP-AE-FMs were used as controls for the experiments. Three parallels were set for each group. The medium was refreshed 3 times a week. Cell viability was examined after 7 and 14 days using alamarBlue assay (Thermo Fisher Scientific, Waltham, MA, USA) and vitality staining, as described earlier [9].

### 4.7. Cell Morphology

DPSCs were cultured in DMEM-F12 media containing 10% FBS, 1% penicillin/streptomycin, and 1% GlutaMAX (all from Thermo Fisher Scientific, Waltham, MA, USA) and were maintained at standard conditions in a humidified 37 °C incubator with 5% CO_2_. Prior to cell seeding, BTCP-AE-FM samples (Ø 12 mm) were sterilized using O_3_, and then UV light (30 min/side). The meshes were laid into the bottom of the well and fixed with sterilized plastic rings. After that, 10^5^ cells were seeded into each well containing fiber mesh, then incubated for 3 days at 37 °C. Cell morphology was studied with SEM (JEOL JSM-IT500HR, Tokyo, Japan). Samples were fixed in 2% glutaraldehyde (Sigma-Aldrich, St. Louis, MO, USA) for 2 h and then in a 1% OsO_4_ solution (Sigma-Aldrich, St. Louis, MO, USA) for 1 h. Samples were dehydrated in ethanol solutions (10%, 30%, 50%, 70%, 80%, 90%, and 100% EtOH) for 15 min for each step, then dried using CO_2_ at a critical point and covered with a gold layer of about 12 nm thickness before microscopic investigation. The accelerating voltage was 15 kV.

### 4.8. Surgical Procedures

Animal surgeries were performed under the authorization of the University of Debrecen Institutional Animal Care and Use Committee (7M/2015/UDCAW). Eighteen (per sample group) adult female Wistar rats weighing 300–350 g (Animalab Kft., Budapest, Hungary) were used in this study. (The sample groups were the crosslinked BTCP-AE-FM and the control group crosslinked FM, n = 6/time point.) After two weeks of quarantine, the animals were randomized into groups of 6. Animals were kept in a conventional animal house at a controlled temperature (26 ± 2 °C) and humidity (55 ± 5%). Artificial lighting was provided in automatically controlled 12-h circadian cycles. The rats were fed *ad libitum* with semi-synthetic feed (VRF1 rodent chow, Akronom Ltd., Budapest, Hungary) and tap water. Rats were anesthetized by 1.5% isoflurane (Forane, AbbVie Ltd., Budapest, Hungary) anesthesia with a dedicated small animal anesthesia device (Tec3 Isoflurane Vaporizer, Eickemeyer Veterinary Equipment, Ghislandi Ltd., Budapest, Hungary). A sagittal skin incision was performed on the scalp, then a matching incision was made on the periosteum as well; soft tissues were separated laterally. An 8 mm diameter full-thickness defect was created in the midline of the parietal region with a trephine bur under saline irrigation (Figure 17a). The bone was then removed with care to avoid dura injury (Figure 17b). The defect in group I was left empty and it remained as the control group; in group II a matching size FM plate and in group III a BTCP-AE-FM plate were implanted at the site of the defect (Figure 17c) [44]. The periosteum and skin were closed in two layers with interrupted resorbable sutures.

Animals were terminated at 1, 3, and 6 months after surgery. Following transcardial perfusion with saline and 4% paraformaldehyde, a vault of the calvaria was then harvested together with the periosteum, keeping 3–5 mm excess bone from the original trepanned edges.

### 4.9. Micro CT Measurement

The structure of the calvaria was determined by a compact desktop micro-CT system, SkyScan 1272 (Bruker, Billerica, MA, USA). The following parameters were used: image pixel size: 10 microns; matrix size: 1344 × 2016 (rows × columns); source and voltage current: 90 kV and 111 µA. An Al 0.5 + Cu 0.038 filter, flat field correction, and geometrical correction were used. The cross-sectional images were reconstructed from tomography projection images by the SkyScan NRecon package (Version: 2.0.4.2). During reconstruction, the post-alignment, beam-hardening correction, ring artifact correction, and smoothing were completed. The output formats were DICOM and BPM.

CTAn software was used in the 2D/3D ROI analyses. The circle ROI designation, thresholding, and 2D and 3D analysis plugins were applied. The ROI mask with a diameter of 8 mm was manually placed on the bone defect. Bone tissue was segmented from the background using a global threshold of 12% of the maximum gray value (global grayscale 0–255).

The 3D visualization was conducted using CTVox software, and the ROI was depicted with blue color coding.

### 4.10. In Vivo PET Imaging Using ^18^F-Sodium Fluoride (^18^F-NaF)

For the imaging of osteoblast activity at the site of the surgery, experimental animals were injected with 8.2 ± 0.4 MBq of ^18^F-NaF via the lateral tail vein 1, 3, and 6 months after the surgery. Fifty minutes after the injection of the radiopharmaceutical, rats were anesthetized by 1.5% isoflurane (Forane, AbbVie Ltd., Budapest, Hungary) anesthesia with a dedicated small animal anesthesia device (Tec3 Isoflurane Vaporizer, Eickemeyer Veterinary Equipment, Ghislandi Ltd., Budapest, Hungary), and static PET scans (20 min acquisition time) were performed using the MiniPET-II camera. PET scanner normalization and random correction were applied to the data. The voxel size was 0.5 × 0.5 × 0.5 mm. After reconstruction (using the standard EM iterative algorithm) the decay-corrected PET images were analyzed using the BrainCad image analysis software (University of Debrecen, Faculty of Medicine, Department of Medical Imaging, Division of Nuclear Medicine and Translational Imaging), and ellipsoidal 3-dimensional volumes of interest (3D VOIs) were manually drawn around the edge of the tissue activity by visual inspection. Radiotracer uptake was expressed in terms of standardized uptake values (SUVs). SUV = [VOI activity (Bq/mL)]/[injected activity (Bq)/animal weight (g)], assuming a density of 1 g/mL.

### 4.11. Histological Analysis

Postfixed calvaria plates were washed for 2 × 30 min in distilled water, then decalcified with EDTA (Osteosoft; Merck, Darmstadt, Germany) for 84 h at 37 °C. Dehydrated samples were then embedded into the paraffin and sectioned at 4 µm thickness in the frontal plane, across the center of the defect, guided by micro-CT reconstructions. Sections were incubated for 4 min in hematoxylin according to Gill (Hematoxylin solution, Gill No. 2; Sigma-Aldrich) and subsequently stained in 95% EtOH medium with eosin for 20 s (Eosin Y alcoholic solution; Leica Biosystems Inc., Richmond, IL, USA).

Microphotographs were captured with an Olympus BX53 conventional light microscope (magnification: 10× and 4× objectives), equipped with an Olympus DP74 color camera (both manufactured by Olympus Ltd., Tokyo, Japan). Diagrams were edited with Adobe Photoshop CS5 (Adobe Systems Inc., San Jose, CA, USA).

### 4.12. Statistical Analysis

Statistical analysis of viability tests was carried out using the Student’s *t*-test to determine the statistical significance of differences between experimental groups (*p* < 0.05-issued to determine significance). GraphPad Prism v8 (GraphPad Software Inc., Boston, MA, USA) was used for the investigations.

## 5. Conclusions

In this work, we demonstrated a novel BTCP-modified aerogel-containing PVA/Ch hybrid mesh with a nanofibrous structure successfully produced by electrospinning. This PVA/Ch hybrid mesh is biocompatible and retains its hydrophilic character, despite partial cross-linking. The hydrophilic nature of the PVA/Ch matrix allows the diffusion of water into the thin BTCP-AE-FM samples, ensuring the release of the inorganic bioactive components (Ca, P, and Si) necessary for bone-healing processes. The prolonged in vivo absorption rate of the BTC-AE-FM may be a consequence of the higher hydrolytic stability of amides over esters, and the material seems to be osteoconductive and promote migration of the osteoblasts, as indicated by the heterotopic bone formation in the inner regions. Our further aim is to improve the biophysical properties of hybrid mesh by increasing the inorganic components using different kinds of additives.

## Figures and Tables

**Figure 1 ijms-24-07562-f001:**
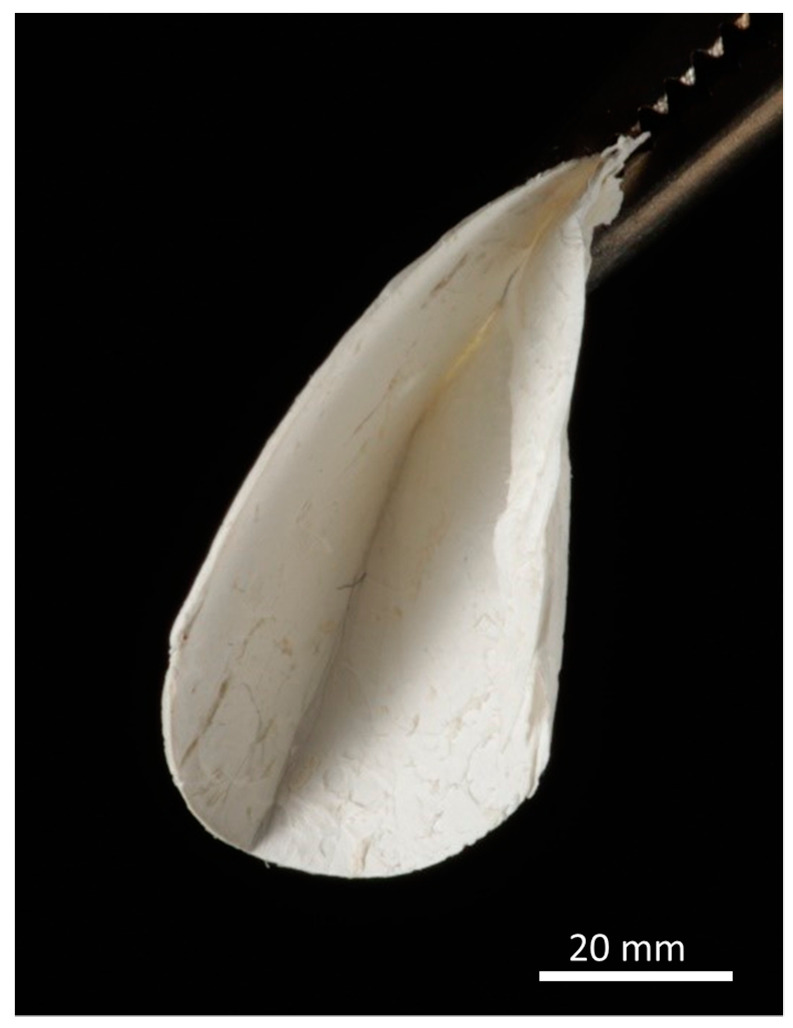
An electrospun BTCP-AE-FM folded in half.

**Figure 2 ijms-24-07562-f002:**
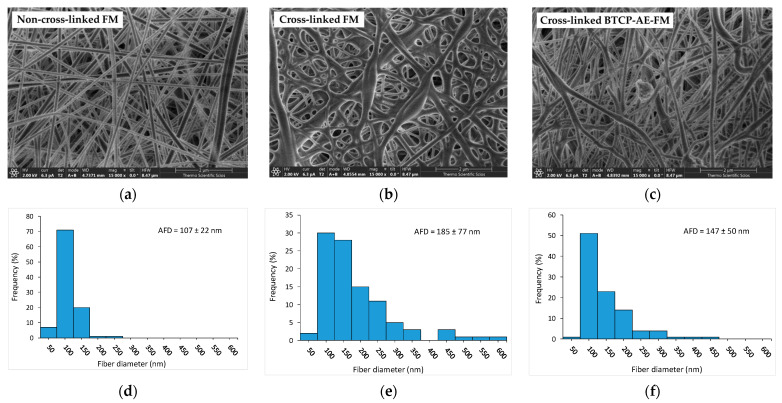
Characterization of the different types of meshes: (**a**) representative SEM micrographs of a non-cross-linked FM, (**b**) a cross-linked FM, and (**c**) a cross-linked BTCP-AE-FM. Their diameter frequency distributions are shown on (**d**–**f**) bar graphs, respectively.

**Figure 3 ijms-24-07562-f003:**
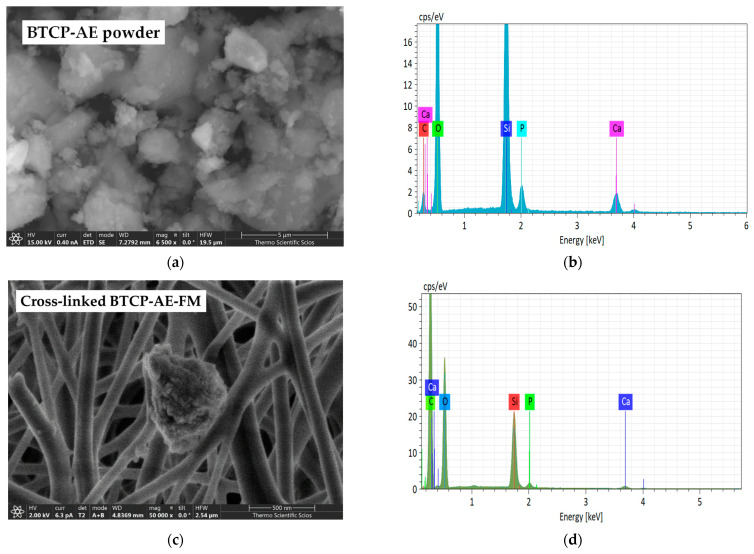
SEM microphotographs of the BTCP-AE powder sample (**a**) and BTCP-AE-FM sample (**c**). EDS analysis of the BTCP-AE inorganic composite powder sample (**b**) and BTCP-AE particle in the composite meshes (**d**).

**Figure 4 ijms-24-07562-f004:**
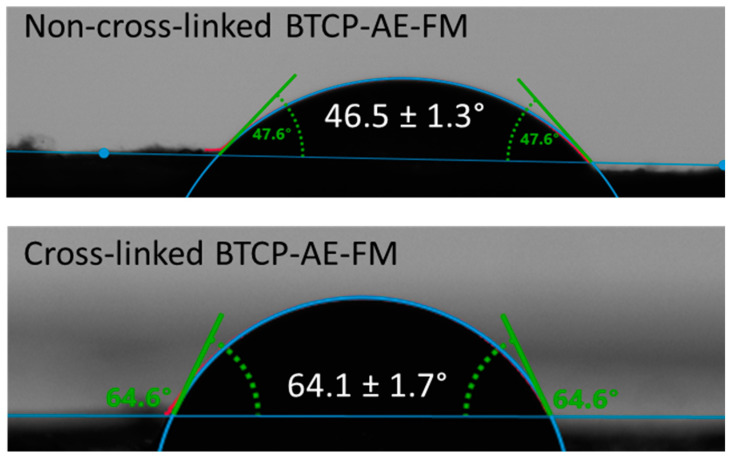
Water contact angles of the BTCP-AE-FMs before and after cross-linking.

**Figure 5 ijms-24-07562-f005:**
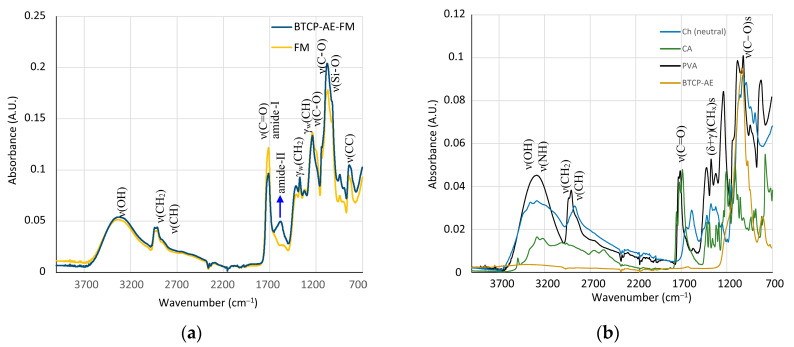
(**a**) IR spectra of the BTCP-AE-FM and FM fibrous meshes. The blue arrow indicates the pronounced presence of amide bonds in the aerogel composite mesh. (**b**) IR spectra of the starting materials showing highly overlapping peaks in the fingerprint region.

**Figure 6 ijms-24-07562-f006:**
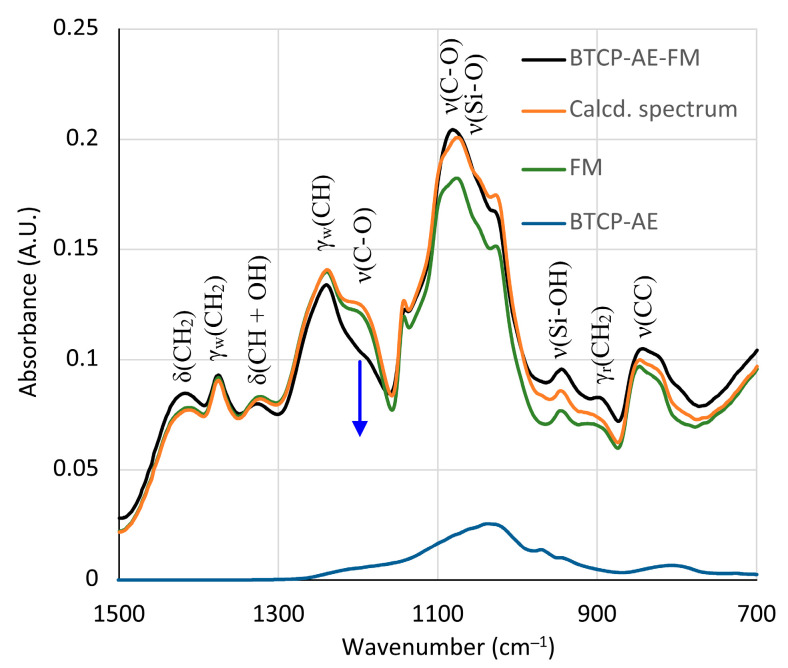
Recorded and best-fit ATR-FTIR spectra of the BTCP-AE-FM nanofiber composite assuming that the spectrum is the linear combination of the spectra of the FM and BTCP-AE. The blue arrow indicates the absorbance drop of the ester group ν(C-O) stretching frequencies.

**Figure 7 ijms-24-07562-f007:**
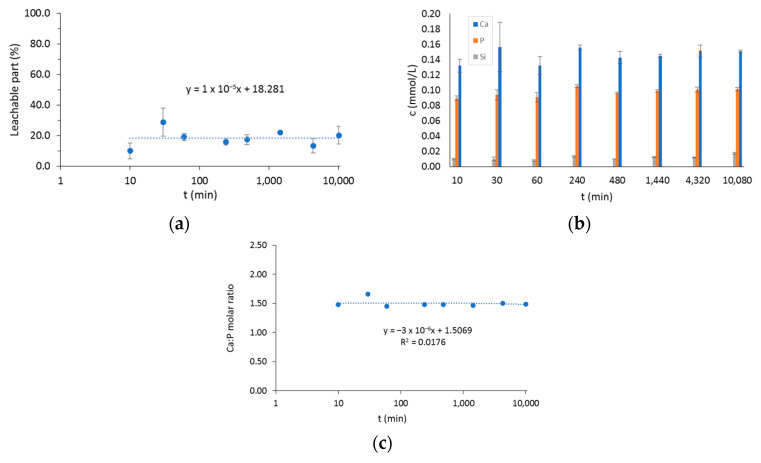
(**a**) The leachable part of the BTCP-AE-FM in water. (**b**) The total amount of Ca, P, and Si ions in water after soaking periods. (**c**) The molar ratio of dissolved Ca and P. The time axis is shown in log scale in plots (**a**,**c**).

**Figure 8 ijms-24-07562-f008:**
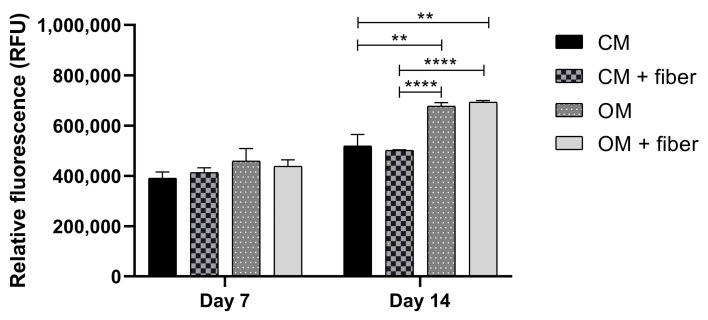
Cell viability assay of DPSC cells: cells were cultured with (CM+ and OM+) or without (CM and OM) BTCP-AE-FM samples for 7 and 14 days. After the incubation period, cell viability was assessed by alamarBlue assay. Values are expressed as sample means; error bars represent the standard deviation (SD) of three parallel measurements. In the *t*-tests, ** denotes *p* < 0.01, and **** denotes *p* < 0.0001. CM: control medium. OM: osteoinductive medium.

**Figure 9 ijms-24-07562-f009:**
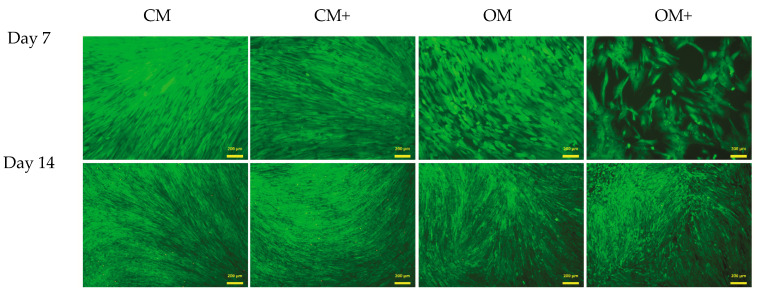
Live–dead assay of DPSCs. Cells were seeded onto 12-well plates and cultured with (CM+ and OM+) or without (CM and OM) BTCP-AE-FM samples for 7 and 14 days. After the incubation period, cells on the different surfaces were co-stained with fluorescein diacetate and propidium iodide.

**Figure 10 ijms-24-07562-f010:**
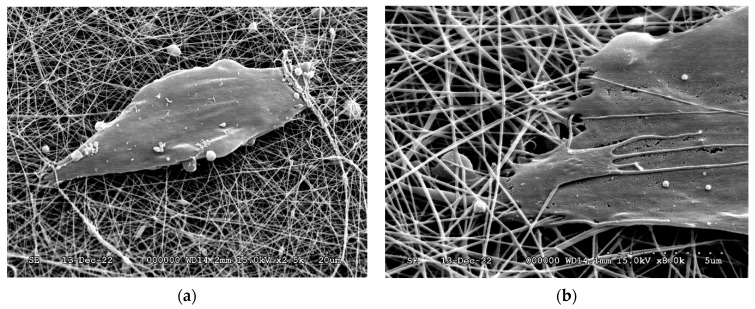
SEM micrographs of DPSCs on the surface of the BTCP-AE-FM (**a**,**b**).

**Figure 11 ijms-24-07562-f011:**
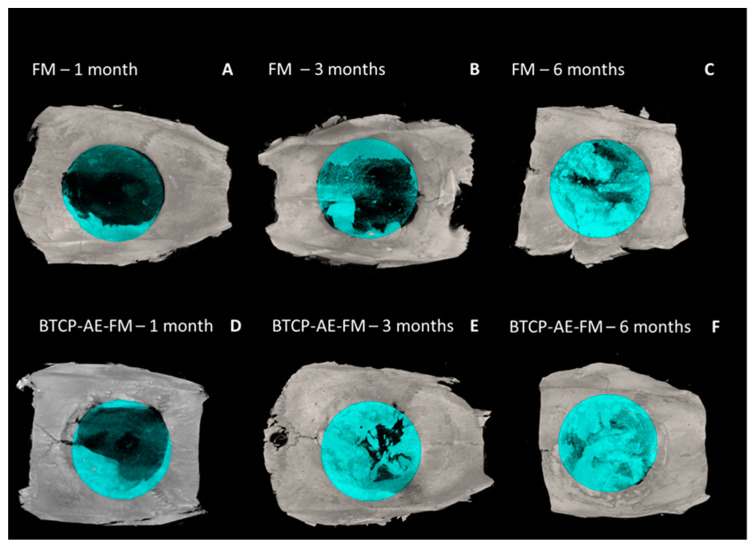
3D surface-rendered micro-CT reconstructions (axial view) with blue-colored ROI mask (8 mm). (**A**–**C**) The FM and (**D**–**F)** BTCP-AE-FM 1–3–6 months.

**Figure 12 ijms-24-07562-f012:**
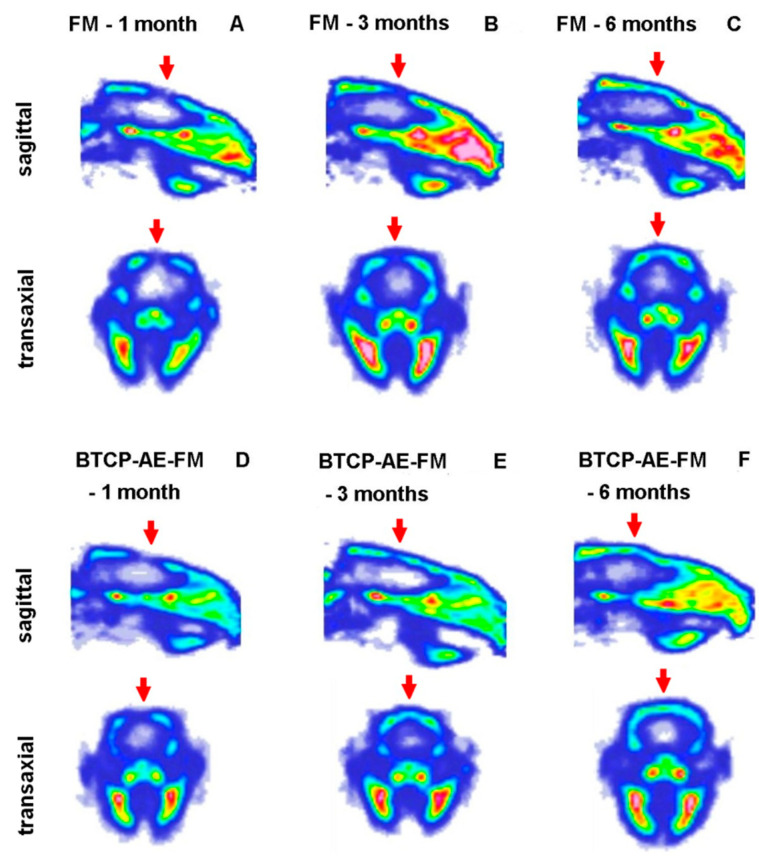
In vivo PET imaging of rat calvaria using [^18^F]fluoride. Representative decay-corrected PET images were obtained 1, 3, and 6 months after surgery and 50 min following intravenous injection of the radiopharmaceutical. (**A**–**C**): sagittal and transaxial PET images of rats implanted with the FM; (**D**–**F**): sagittal and transaxial PET images of rats implanted with the BTCP-AE-FM. Red arrows: area of the cranial trepanation surgery.

**Figure 13 ijms-24-07562-f013:**
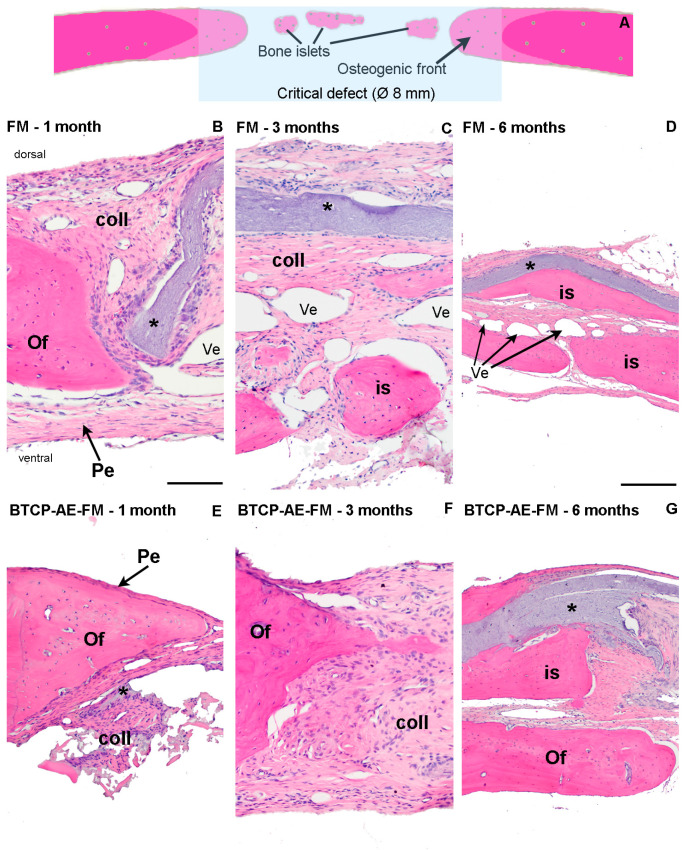
Histological analysis of bone remodeling in HE-stained rat calvaria plates. (**A**) The schematic diagram explains multifocal bone remodeling during calvarial defect repair. Progress of reossification was demonstrated with HE staining in the critical defect, applying polyvinyl alcohol and chitosan blended fibrous mesh (FM) (**B**–**D**) and inorganic composite modified polyvinyl alcohol and chitosan hybrid fibrous mesh (BTCP-AE-FM) (**E**–**G**) implants. Magnification: 10× obj (**B**,**C**,**E**,**F**); 4× obj (**D**,**G**). Pe: periosteum. Of: osteogenic front. is: bone islets. Coll: collagenous tissue. Ves: blood vessels. *: implanted fibrous mesh. Scalebar: 0.5 mm (**B**,**C**,**E**,**F**); 1 mm (**D**,**G**).

**Figure 14 ijms-24-07562-f014:**
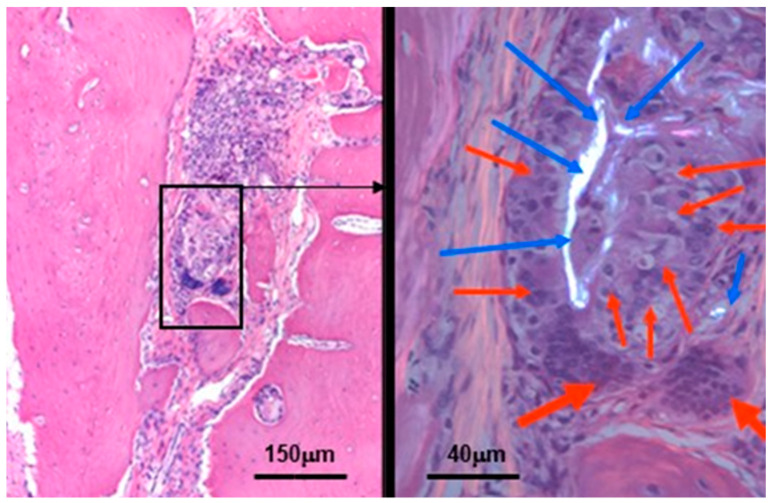
Residual silica compounds following mesoporous BTCP-AE insertion into the calvaria bone defect 6 months after treatment. The hematoxylin-eosin (HE) stained image (**left**) shows sub-total re-ossification of the defect at this time, highlighted by the solid hypocellular pink tissue. However, the center of the picture still mainly exhibits an inflammatory fibrous non-ossified region with the presence of a foreign body giant cell granulomatous reaction in association with the presence of non-metabolized silica crystal particle remnants (within the black frame). This region is magnified (630×) in the right-hand side image, which was photographed in a polarizing microscope to confirm the presence of crystalloid foreign particles (the birefringent bright white materials, indicated by the blue arrows) surrounded by the epithelioid-activated macrophages (thin red arrows) and the multinuclear (foreign body) giant cells (thick red arrows) [32].

**Figure 15 ijms-24-07562-f015:**
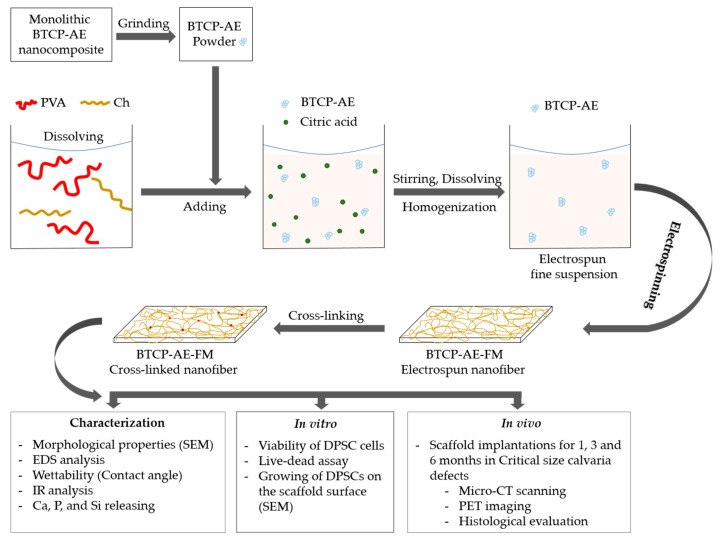
Workflow of the fabrication, characterization, and in vitro and in vivo investigations of chemically cross-linked biodegradable BTCP-AE-FMs.

**Figure 16 ijms-24-07562-f016:**
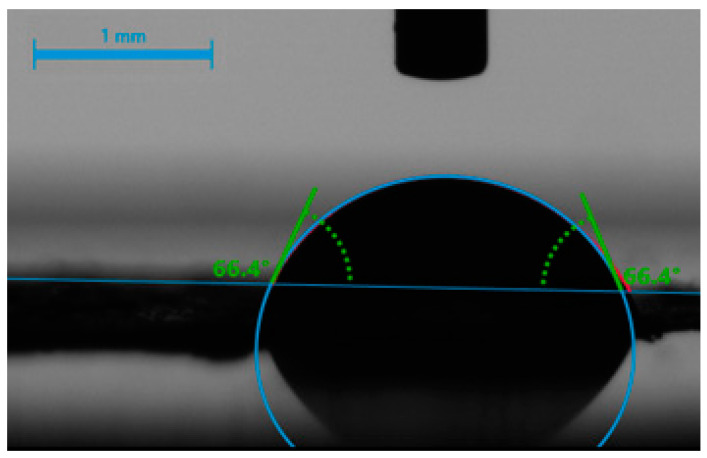
Contact angle measurement of a water droplet (2 µL) on a BTCP-AE-FM sample.

**Figure 17 ijms-24-07562-f017:**
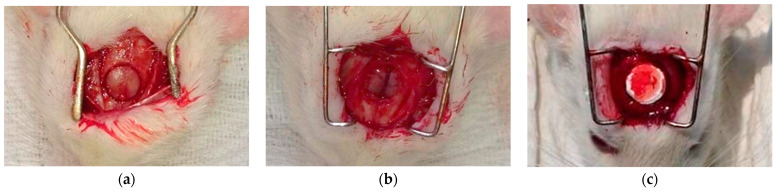
The surgical process for the critical size calvarial defects in rats. (**a**) Preparation of the 8 mm diameter full-thickness defect. (**b**) Surgical area after bone removal. (**c**) Implanted BTCP-AE-FM plate in the defective site.

**Table 1 ijms-24-07562-t001:** Characteristic IR absorptions and plausible assignations of peaks of the BTCP-AE-FM nanofiber composite.

Wavenumber (cm^−1^)	Assignation	Wavenumber (cm^−1^)	Assignation
3317.5 m br	ν(OH)	1328.3	δ(CH + OH)
2939.5 w	ν(CH_2_)	1239.5	γ_w_(CH)
2913.0 w	ν(CH_2_)	1200 sh	ν(C-O)
2873.5 vw sh	ν(CH)	1081.4	ν(C-O)
1713.0	ν(C=O), amide-I	944.5	
1585.2	amide-II	899.2	γ_r_(CH_2_)
1417.0	δ(CH_2_)	842.8	ν(CC)
1375.0	γ_w_(CH_2_)		

**Table 2 ijms-24-07562-t002:** Quantitative SUV analysis of the decay-corrected PET images obtained 1, 3, and 6 months after the surgery and 50 min after the intravenous injection of [^18^F]fluoride. n = 6/time point.

	1 Month SUV_mean_	3 Months SUV_mean_	6 Months SUV_mean_
FM implant	5.41 ± 1.12	6.84 ± 1.01	7.40 ± 1.03
BTCP-AE-FM implant	5.53 ± 1.08	8.48 ± 0.94	10.72 ± 1.11

## Data Availability

The raw data supporting the conclusions of this article will be made available by the authors, without undue reservation.

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
