# Peer review of "β-Tricalcium Phosphate-Modified Aerogel Containing PVA/Chitosan Hybrid Nanospun Scaffolds for Bone Regeneration"

_ijms, 2023, doi:10.3390/ijms24087562_

Round 1
Reviewer 1 Report
This paper is well written and reports an interesting study conducted with adequate methods. Results are clearly presented and well discussed. Characterizations are comprehensive.
Some minor suggestions are given for further improvement.
1. This review paper on nanofibrous constructs deserves to be cited:
Electrospun nanofibers for improved angiogenesis: Promises for tissue engineering applications. Nanomaterials 2020;10:1609
2. Some key characteristics of BTCP powders, e.g. morphology by SEM and particle size distribution, should be reported.
3. XRD of BTCP should be reported, too.
4. Have the authors acquired any evidence of mechanical properties of the scaffolds?
Reviewer 2 Report
This manuscript reported the development of novel electrospun hybrid composite scaffolds, and further explored their performances for bone tissue engineering application. This work is interesting, and pretty attractive. Some major revisions are suggested to be conducted before publication.
1. Title is a little bit misleading. The full name of BTCP should be given. What’s meaning of “Nanospun”? The professional term should be used to increase the readability.
2. Abstract should be presented in a better and clear way. For instance, the authors paid more attention on what they did. Instead, some more descriptions should be added to introduce the results and conclusions. In addition, some important result data are suggested to be presented in this section.
3. Please state the reasons why both PVA and Ch were selected in this study. What are the merits and demerits of PVA and Ch, compared with the other commonly used biopolymers.
4. The merits of electrospinning technique should be further outlined, and some recent works about the innovative electrospinning like https://doi.org/10.1016/j.eurpolymj.2023.111863 and https://doi.org/10.1021/acsami.1c24131 are suggested to be discussed.
5. How did the author select the polymeric concentration and electrospinning parameters for electrospinning? Do they conduct any preliminary experiments?
6. The scale bar is missing in Figure 4, which should be added. Please state the reasons why the crosslinking process could improve the fiber diameter of FM in Figure 5. In addition, Figure 6c is not clear enough, and the images with high quality should be utilized. Please justify the reasons why the crosslinking process could improve the water contact angle in Figure 7.
7. Some important characteristic peaks should be labeled and highlighted in Figure 8 and 9. Did it have any significant difference in Figure 11. Statistical analysis should be conducted for biological tests. Scale bars are missing in Figure 12.
8. The grammar and writing should be improved in the whole manuscript.
Reviewer 3 Report
1. The abstract should be broadened to give additional quantitative results.
2. Please conclude your abstract with a "take-home" message.
3. Rearrange the keywords so that they are in alphabetical order.
4. Abbreviation as a keyword is not recommended and encouraged to be changed to become a stand for its abbreviation.
5. It is unclear whether the author's something new in this work. According to the evaluation, several published literature by other researchers in the past adequately explain the issues you made in the present paper. Please be careful to highlight in the introduction section anything really innovative in this work.
6. Previous literature related needs to explain in the introduction section consisting of their work, their novelty, and their limitations to show the research gaps that intend to be filled in the present work.
7. Sometimes in the entire manuscript, the authors created paragraphs with just one or two phrases, which made the explanation difficult to understand. To make their explanation a full paragraph, the authors must expand it. It is advised to use at least three sentences in a paragraph, with one acting as the primary sentence and the other two as supporting phrases.
8. The current submission needs to make proper editing in the objective of the present work in the last paragraph of the introduction section.
9. In line 86, in study objective, the authors stated “….scaffolds during bone regeneration,”. However, in the introduction section, the authors does not explain any concept and role of scaffold in bone regeneration (and tissue engineering). The authors needs to explain this important topic at least one paragraph (or minimum one sentence). Also, relevant reference needs to adopted to support the explanation as follows: The Effect of Tortuosity on Permeability of Porous Scaffold. Biomedicines 2023, 11, 427. https://doi.org/10.3390/biomedicines11020427
Round 2
Reviewer 2 Report
The authors have addressed the reviewer's comments well.
Reviewer 3 Report
Address following comments in the stage.
1. Please provide an additional figure in the introduction section in related submission work to improve the reader's understanding.
2. In order to improve the reader's understanding of the materials and methods section simpler, the authors could provide figures that clarify the workflow of the current study rather than only the predominant text as it currently appears.
3. Please explain more clearly the basis of patient selection since the present form was insufficient. Is any standard, procedure, or protocol used? The involved patient is also very small and heterogeneous without real group control. It is urgent since impacting the obtained results would lead to biased analysis. Fatal flaws that need to be addressed after revision.
4. In line 105-112, the explanation of scaffold needs additional updated reference to refer as follows: Level of Activity Changes Increases the Fatigue Life of the Porous Magnesium Scaffold, as Observed in Dynamic Immersion Tests, over Time. Sustainability 2023, 15, 823. https://doi.org/10.3390/su15010823
5. Additional information about tools used, such as the maker, country, and specification, should be included.
6. Important information that must be included in the publication refers to the error and tolerance of the experimental equipment used in this inquiry.
7. Results must be compared to similar past research.
8. Because the current quality is not appropriate, the authors must improve their discussion to add more depth.
9. The limitation of the present submission needs to be added at the end of the discussion section before entering the conclusion section.
10. Elaborate the conclusion as a form paragraph, not point by point as in the present form.
11. The conclusion section needs to explain further research.
12. Literature from the last five years should be enriched to reference. MDPI reference is strongly recommended.
13. The authors need to reduce their level of self-citation with using literature that not authored by the present authors in the current submission as a reference.
14. Due to grammatical mistakes and English style, English has to be proofread.
15. After revision, provide a graphical abstract for submission.
Round 3
Reviewer 3 Report
Nice effort to the authors in this stage. Minor comments given to the manuscript. Since the presents manuscript evaluate in in vivo and in vitro sides, but not conducted in silico study, encouraging to explain potential further study performing in silico. In silico brings several advantages such as lower cost and faster results compared to in vivo and in vitro. Refer the relevant reference to address this issue as follows: In Silico Contact Pressure of Metal-on-Metal Total Hip Implant with Different Materials Subjected to Gait Loading. Metals (Basel). 2022, 12, 1241. https://doi.org/10.3390/met12081241
Round 4
Reviewer 3 Report
In the moment, I found that the authors do not discuss the urgency of biomechanical and biocompability aspect of materials selection of implant materials. Please explain it in brief in the introduction section to improve the understanding for the reader. Authors also encouraged to incorporated relevant reference as follows: Polycrystalline Diamond as a Potential Material for the Hard-on-Hard Bearing of Total Hip Prosthesis: Von Mises Stress Analysis. Biomedicines 2023, 11, 951. https://doi.org/10.3390/biomedicines11030951